# The Use of High-Quality Watermelon Seedlings Is Prerequisite to Limit the Transplanting Shock and Achieve Yield Earliness

**Filippos Bantis** [1,*] and **Athanasios Koukounaras** [2]

1 Department of Agriculture, University of Patras, 30200 Messolonghi, Greece
2 Department of Horticulture, Aristotle University, 54124 Thessaloniki, Greece; thankou@agro.auth.gr
* Correspondence: fbantis@upatras.gr

**Abstract:** One of the most important crops of the Mediterranean, watermelon (*Citrullus lanatus*), is almost exclusively established through seedlings. For many years, agronomists, nurseries, and farmers have aimed to produce and use high-quality seedlings for better growth in the field. However, seedling quality has not been examined as to what defines the subsequent plant, flower, and fruit development, and to what extent. Our aim was to test whether different seedling qualities labeled as "*Optimum*", "*Acceptable*", or "*Not Acceptable*" for cultivation actually perform variably in terms of vegetative, flowering, and fruit development, as well as fruit quality after a full growing cycle in the field. Vegetative growth (stem diameter, plant area, and leaf number) was evaluated until flowering and was enhanced for *Optimum* plants. The flowering of *Not Acceptable* plants started two days later, while *Optimum* plants retained a greater number of female flowers throughout the two-week blooming evaluation. Most importantly, *Optimum* plants developed mature fruits four and six days faster than the *Acceptable* and *Not Acceptable* ones, respectively, showing considerable yield earliness. The photosynthetic mechanism, as well as fruit morphology and phytochemical content, were not affected by quality categories. Overall, indeed it is important to use high-quality seedlings to achieve yield earliness of watermelon fruits.

**Keywords:** *Citrullus lanatus*; nursery; transplantation; morphology; photosynthetic mechanism; flowering; crop production; antioxidants





## 1. Introduction

Nowadays, the most economically important vegetable species, such as the ones belonging to the Cucurbitaceae, Solanaceae, Brassicaceae, and Asteraceae families, are widely established in the field with seedlings. Compared to direct sowing, seedlings provide a plethora of benefits, including seed saving, reduced plant losses, higher homogeneity within the field, better crop scheduling, and yield earliness [1]. Moreover, grafted seedlings also provide resistance to abiotic and biotic stressors [2]. The production and establishment of high-quality seedlings is of utmost importance for growers to enjoy rapid and vigorous plant development, extensive flowering, and good yield. This necessity led to the development of image-processing software for the determination of seedling quality [3]. Such systems require the determination of quality indices which may be different across species. For example, a study involving grafted watermelon showed that seedlings of high quality could be distinguished by developmental parameters such as stem diameter, leaf area, and shoot and root dry masses, among others [4]. From such parameter evaluations, it is possible to predict with relative reliability the subsequent growth potential of seedlings upon their transplantation in the field [5]. Experienced growers evaluate the seedling development visually, probably by subconsciously taking into consideration the abovementioned characteristics. Mechanical engineering could provide help in this direction with cameras and specialized software that selects the best seedlings for further cultivation and the low-quality seedlings for discarding. However, the most important time

for actual seedling evaluation is during the first days after transplantation, when plants need to mitigate the transplant shock and thrive. Transplant shock can be defined as the mortality or decelerated plant growth soon after transplanting [6]. An increased ratio of root-to-shoot growth has been reported as an important factor in limiting transplant shock of muskmelon transplants in diverse environments [7].

Watermelon (*Citrullus lanatus* L.) is a popular and delicious crop mainly cultivated in warm subtropical regions throughout the world. The overall export value of 2021 reached EUR 2.02 billion, which was 24.9% greater than in 2017, showing an increasing tendency for its consumption on a worldwide scale. The exports of European countries, in particular, reached almost half (47.4%) of the worldwide export value (FAOSTAT Database, 2022).

The crop is established in the field with seedlings. Worldwide, it is very common that watermelon seedlings are grafted onto cucurbits such as squash or even other watermelon hybrids in order to alleviate the negative impacts of considerable stress factors (i.e., extreme temperature, heavy metals, nematodes etc.) which deteriorate crop yield and quality [8,9]. In Greece and other major watermelon producer countries, watermelon is established in the field through grafted seedlings at a high percentage: over 90% of the total seedlings established [2,10], in low tunnels in the early spring to promote yield earliness.

Grafted watermelon seedlings are commonly produced by the farmers themselves or by commercial nurseries, which are rapidly expanding [11]. Commercial nurseries typically achieve considerably higher quality compared to individual farmers in terms of seedling sturdiness and overall development. Moreover, they typically produce a smaller amount of lower-quality seedlings, which are supposedly not fit for cultivation, and thus, are discarded. In addition, since watermelon is a highly seasonal crop, nurseries usually struggle with land availability during seedling production, making quality categorization even more important to achieve maximum efficiency and reduce operating costs. For example, a study of our group [12] showed that lower-quality scion and rootstock seedlings must be avoided before grafting in order to significantly increase the productivity of a large-scale commercial nursery.

To this day, it is not recorded, thus, unknown, whether grafted watermelon seedling quality before transplantation directly corresponds to the transplanting shock limitation and the overall plant development and yield in the field. Our research hypothesis was that grafted seedlings of higher quality should perform better compared with the ones of lower quality, both during the first crucial days after transplanting, and with respect to the crop earliness and agronomic attributes. Different quality categories for grafted watermelon seedlings were already described in another publication of our group [4].

To this end, we cultivated grafted seedlings from different quality groups labeled as "*Optimum*", "*Acceptable*", or "*Not Acceptable*". Our aim was to test whether these quality groups perform variably in terms of plant growth, flowering, yield, and fruit quality after a full growing cycle in the field. Our findings can help watermelon seedling producers and farmers to select the best option of quality categories to be transplanted in the field.

## 2. Materials and Methods

### 2.1. Plant Material and Quality Categories

The experiment took place in 2021 at the farm (N 40.536; E 22.995) of Aristotle University of Thessaloniki, Greece. The grafted watermelon seedlings (scion hybrid "Celine F1"; rootstock *Cucurbita maxima* × *C. moschata* hybrid "TZ-148") were produced and provided by a commercial nursery (Agris S.A. in Kleidi, Imathia, Greece). A detailed description of the grafted watermelon seedlings' production process can be found in recent publications of our group [10].

The stage of acclimatization was conducted in a plastic Venlo-type greenhouse for 14 days. Here, minimum night temperature was above 21.5 °C, and supplemental lighting was provided with high-pressure sodium lamps providing $60 \pm 10$ µmol m$^{-2}$ s$^{-1}$ photosynthetic photon flux density for an 18 h photoperiod. Relative humidity was 60–80% and the plants were sub-fertigated. After acclimatization and before transplantation in the



field, the seedlings were grouped into quality categories with assistance from experienced personnel. Specifically, the seedlings were characterised as "*Optimum*", "*Acceptable*" (i.e., acceptable for the market), and "*Not Acceptable*" (i.e., unacceptable for the market). It should be mentioned that the groups were labeled (*Optimum*, *Acceptable*, and *Not Acceptable*) according to previous experiments as well as several observations from the company's experienced personnel. The macroscopic evaluation took into account the leaf area, the colour, the root system, and the overall seedling vigor, which were examined in previous experiments [4,12]. At the end of acclimatization and before transplanting in the field, *Optimum*, *Acceptable*, and *Not Acceptable* seedlings had a leaf area of about 50, 35, and 30 cm$^2$, respectively. Moreover, stem diameter was about 4.65, 4.50, and 4.35 mm for *Optimum*, *Acceptable*, and *Not Acceptable* seedlings, respectively. *Optimum* and *Acceptable* seedlings had 3 true leaves and a smaller fourth leaf, while *Not Acceptable* seedlings had 3 true leaves. The root systems of *Optimum* and *Acceptable* seedlings filled about 1/3 and 1/4 of the substrate in the plug trays, respectively. In addition, for *Not Acceptable* seedlings, only a few root hairs were protruding from the sides of the cell block. Leaves and cotyledons of all categories had deep green colour, while all seedlings were healthy.

### 2.2. Cultivation in the Field

The field soil is of moderate to heavy SCL (sandy clay loam), with 1.8% organic matter, 7.2 pH, and 0.80 mS/cm electrical conductivity. Soil N, P, K, Mg, Fe, Zn, Mn, Cu, and B were 53, 46, 392, 623, 11, 2.9, 32, 1.5, and 0.6 ppm, respectively. Typical cultivation practices took place before and during the experiment, such as plowing, fertilization, irrigation, and weed and pathogen control. For fertilization, a 20-5-20 (NPK) including 3 units of Mg was applied at 500 kg per hectare, while the crops were also fertilized with 13.5-0-46 $KNO_3$ twice during the experiment.

Before transplantation in the field, 16 seedlings from each quality category/treatment were grouped into four subgroups containing four seedlings. Then, the seedlings were transplanted in two cultivation rows at distances of 3 m between rows and 1 m within rows. At the end, each row consisted of two subgroups from every quality category. The plants were arranged in an RCBD (randomized complete block design), while transplantation took place on 2 June 2021.

Meteorological data were obtained from a station located 2.9 km (N 40.562; E 22.995) from our experimental site. Briefly, mean air temperatures in June, July, and August (up to the harvest day) were 24.4, 28.6, and 30.2 °C, respectively. Precipitation was very low at 31, 9, and 2 mm in June, July, and August, respectively.

### 2.3. Measurements and Analyses

As a general rule, measurements and analyses were conducted in the middle plants of each subgroup. In the initial phase of the experiment and before flowering, stem diameter, plant area, and leaf number were measured for the first two weeks after transplanting, on days after transplantation (DAT) 0, 7, and 14. Stem diameter was measured at the plant base, 2 cm above the soil using a digital caliper. The measurement of plant area was selected instead of leaf area since the latter is a destructive measurement while our crop was ongoing. For plant area evaluation, images were taken directly on top of each plant. The images were analysed using WinRHIZO Pro software (Regent Instruments Inc., Quebec, Quebec City, Canada) which identified and separated pixels of different colouration. Within each image, the surface area of non-green parts (e.g., soil, soil cover etc.) was excluded, while the surface area of green plant parts was included in the evaluation. On DAT 14, the maximum quantum yield of primary photochemistry (Fv/Fm) was determined with a chlorophyll fluorometer (Pocket PEA, Hansatech, King's Lynn, UK) on dark-adapted leaves, while relative chlorophyll content was measured with a chlorophyll content meter (CCM-200, Opti-Sciences, Hudson, NH, USA).

Watermelon is monoecious, and male flowers bloom a few days earlier than female flowers. In this experiment, male flowers started to bloom on DAT 21, while female flowers

started to bloom on DAT 24. Therefore, on DAT 24, we started to measure and tag the female flowers with plastic labels every two days until after peak flowering on DAT 36 (i.e., on DAT 24, 26, 28, 30, 32, 24, and 36). The labels were placed with the purpose of identifying the flowering dates and quality categories of the produced fruits at the end of the experiment. The average and sum of female flowers were determined on each date.

For watermelon hybrids grafted onto *C. maxima* × *C. moschata* hybrid TZ-148, such as in our case, 40–45 days is considered as the optimum length of time time from flowering to harvest, depending on the environmental conditions. Moreover, total soluble solids above 11 °Brix is also a safe indicator for watermelon fruit maturity. During our experiment, conditions were exceptionally hot, and fruits were already mature after 40 days from flowering. Therefore, at the end of the experiment, after 40 days from each date of flowering, fruits were harvested separately for each date. To be more precise, fruits tagged on DAT 26 were harvested on DAT 66 after 40 days from flowering, fruits tagged on DAT 28 were harvested on DAT 68 after 40 days from flowering, etc. Female flowers on DAT 24 were only few in number and did not lead to produced fruits. Yield and fruit numbers are presented for each date of harvest.

Upon harvest, three fully developed fruits of similar weight (about 8 kg) from each quality category were selected for further determination. Regardless of quality category, fruits weighed about 7–9 kg, which is typical for the specific hybrid; thus, we opted to analyze average fruits for maximum uniformity among quality categories. In addition, fruits of different sizes would have different water amounts and probably similar absolute biochemical compound content, leading to different compound dilutions. Initially, fruit length and width were measured with a ruler, while rind thickness was measured after the fruits were cut horizontally.

During biochemical analyses, the fruit flesh was homogenized, and the light absorbance of the coloured products was measured through a spectrophotometer (Shimadzu Scientific Instruments, Columbia, MD, USA). For antioxidant capacity determination, the method of ferric reducing antioxidant power (FRAP) [13] was conducted, by which the homogenized fruit samples were extracted using 80% aqueous methanol. Subsequently, the working solution was prepared, with $FeCl_3$, TPTZ, and $CH_3COONa$ pH 3.6 added to the samples, and incubated (4 min at 37 °C), while the light absorbance of the product was determined at 593 nm.

For total phenolic content determination [14], the homogenized fruit samples were also extracted using 80% aqueous methanol. The coloured product was prepared after the addition of 10% Folin–Ciocalteu's reagent and 7.5% sodium carbonate in the methanolic extract, while the light absorbance of the product was determined at 760 nm after incubation (5 min at 50 °C).

Determination of total carotenoid and lycopene contents was also conducted [15] in the homogenized fruit samples, which were extracted with 80% ethanol, while the light absorbance of the product was determined at 503 and 445 nm. Finally, total soluble solids (°Brix) were measured using a refractometer (PAL-α, Atago, Tokyo, Japan).

Statistical analysis (ANOVA) was conducted through SPSS software (SPSS 23.0, IBM Corp., Armonk, NY, USA). Post-hoc analysis was conducted at significance level a = 0.05 using the method Scott–Knott [16] (StatsDirect, Ltd., Birkenhead, UK). The Scott–Knott method presents important and unique characteristics by which the grouping results do not overlap. This is critical to obtain values that separate the three quality categories without overlapping results between them [17].

## 3. Results and Discussion

After transplantation in the field and up to the first male flower blooming on DAT 19, we evaluated a number of valuable vegetative parameters. Stem diameter, a valuable index of watermelon [4], cucumber [18], and Solanaceae crops' [19] quality, was similar on DAT 0 and DAT 7 at all quality categories, though a tendency for increased values was visible in *Optimum* plants. Nevertheless, on DAT 14, stem diameter was indeed significantly

greater in *Optimum* compared to *Acceptable* and *Not Acceptable* plants (Figure 1A). A thicker stem is indicative of the plant's ability to transport a larger amount of water and nutrients through its vascular system, which is particularly important for fast-growing crops such as watermelon during the warm months.

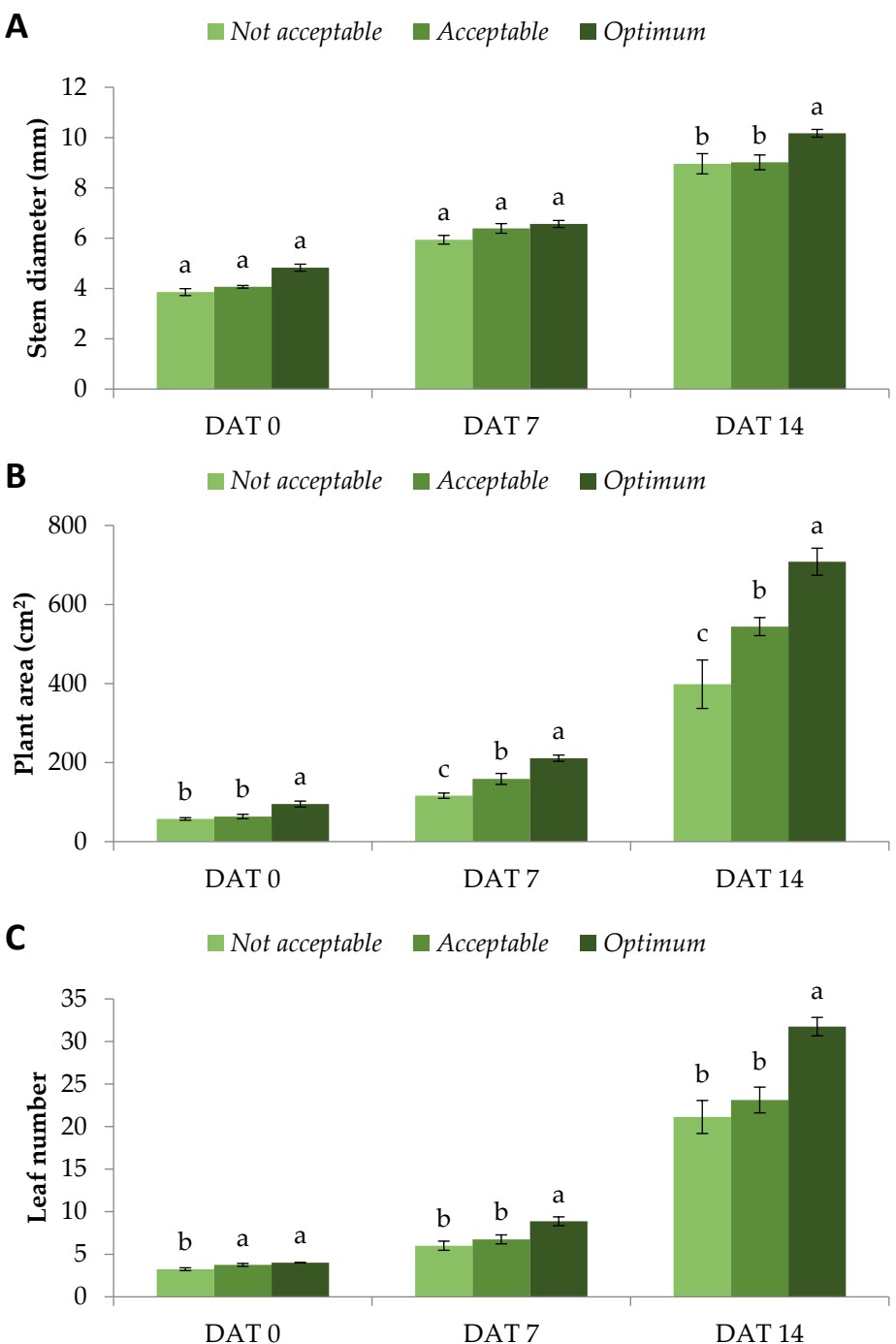

**Figure 1.** (**A**) Stem diameter, (**B**) plant area, and (**C**) leaf number of watermelon plants from different quality categories at 0, 7, and 14 days after transplanting (DAT). Within a DAT, average values (*n* = 8; ±SE) followed by different letters are significantly different ($\alpha < 0.05$).

An even clearer increment was observed for the plant area. A similar method of area determination was described for tomato, cucumber, aubergine, and pepper seedlings, with an accuracy of over 95% [20]. Specifically, from DAT 0 and up to DAT 14, *Optimum* plants produced significantly larger plant area, +30% and +78% compared with the *Acceptable* and

*Not Acceptable* quality categories, respectively. Moreover, on DAT 7 and DAT 14, *Acceptable* plants also showed greater values compared with the *Not Acceptable* ones (Figure 1B). The abovementioned observations are also backed up by leaf number, which was significantly decreased for *Not Acceptable* plants throughout the vegetative evaluation period (DAT 0 to DAT 14) compared with the *Optimum* quality category, which showed 37% greater values. In addition, the *Optimum* plant also had significantly more leaves compared with the *Acceptable* ones on DAT 7 (+48%) and DAT 14 (+50%) (Figure 1C). Even from the day of transplantation on DAT 0, both plant area and leaf number were considerably different between the quality categories, which shows that the seedlings to be transplanted were meticulously selected. Until DAT 7, the plant areas of *Optimum*, *Acceptable*, and *Not Acceptable* plants were +122%, +150%, and +102% greater compared with DAT 0, respectively. The respective differences between DAT 14 and DAT 7 were +235%, +243%, and +243%. In addition, the leaf numbers of *Optimum*, *Acceptable*, and *Not Acceptable* plants were +122%, +80%, and +85% greater compared with DAT 0, respectively. The respective differences between DAT 14 and DAT 7 were +258%, +243%, and +252%. The abovementioned percentage differences show that the growth rate of *Optimum* and *Acceptable* plants during the first week in the field was considerably greater compared with the *Not Acceptable* ones. The latter quality category only managed to match the pace during the second week after transplanting, but the plants already lagged considerably. This growth deceleration of *Not Acceptable* plants can be attributed to the inferior root system during transplantation [4], which further deteriorated the transplanting shock suffered by the plants.

Moreover, we attempted to evaluate the photosynthetic mechanism on DAT 14. Our results show that the relative chlorophyll content and Fv/Fm were not significantly different among the different quality categories. The latter was within the values (i.e., 0.78–0.86) reported by Björkman and Demmig [21] as expected to be measured in non-stressed plants, indicating that the crops were growing healthily and in a regular manner.

The momentum gained during the rapid vegetative growth was also retained during flowering. As mentioned above, male flowers started blooming at DAT 19. Female flowering was initiated on DAT 24 for *Optimum* and *Acceptable* plants, with the former showing more female flowers. *Not Acceptable* plants started female flowering two days later, on DAT 26. On DAT 26 and DAT 28, *Optimum* female flowers bloomed at a significantly greater pace compared to the other quality categories. From DAT 30 and onwards, daily female flowering was similar for all quality categories (Figure 2A).

The abovementioned momentum led to a significantly greater sum female flower number of *Optimum* plants from DAT 24 up to DAT 34, while on DAT 36, there was also a strong tendency towards the same quality category. Moreover, *Acceptable* plants showed significantly greater sum female flower numbers compared to *Not Acceptable* from DAT 26 up to DAT 30 (Figure 2B). Flowering is a complex process involving molecular and environmental signals leading to several pathways regulated by microRNAs, such as miR156 [22] and miR172. These also interact with a plethora of endogenous and exogenous factors, which in turn regulate the flowering time of plants [23]. The authors concluded that enhancing the understanding of flowering regulation is critical in order to increase crop yield and biomass and to shorten the vegetative period of many crops.

After transplantation in the field, seedlings undergo the so-called "transplantation shock", which limits plant growth and development for a period. The length of this period, along with the intensity of the transplantation shock, depends on the species and its ability to rapidly increase its root system and subsequently enhance its vegetative growth. Before transplantation, *Not acceptable* seedlings were characterized by a considerably more compact root system compared with the other quality categories. Upon visual evaluation, the root system of *Not acceptable* seedlings was barely protruding from the sides of the cell block in the plug trays. On the contrary, the root systems of *Optimum* and *Acceptable* seedlings filled up about 1/3 and 1/4 of the cell block, respectively, and were clearly protruding from the sides. The denser root systems of *Optimum* (mainly) and *Acceptable* (secondarily) seedlings allowed for the plants to absorb a larger amount of water and nutrients and accelerate their

establishment in the field compared with the *Not acceptable* ones. The above led to general slower development of the latter, including decelerated flowering.

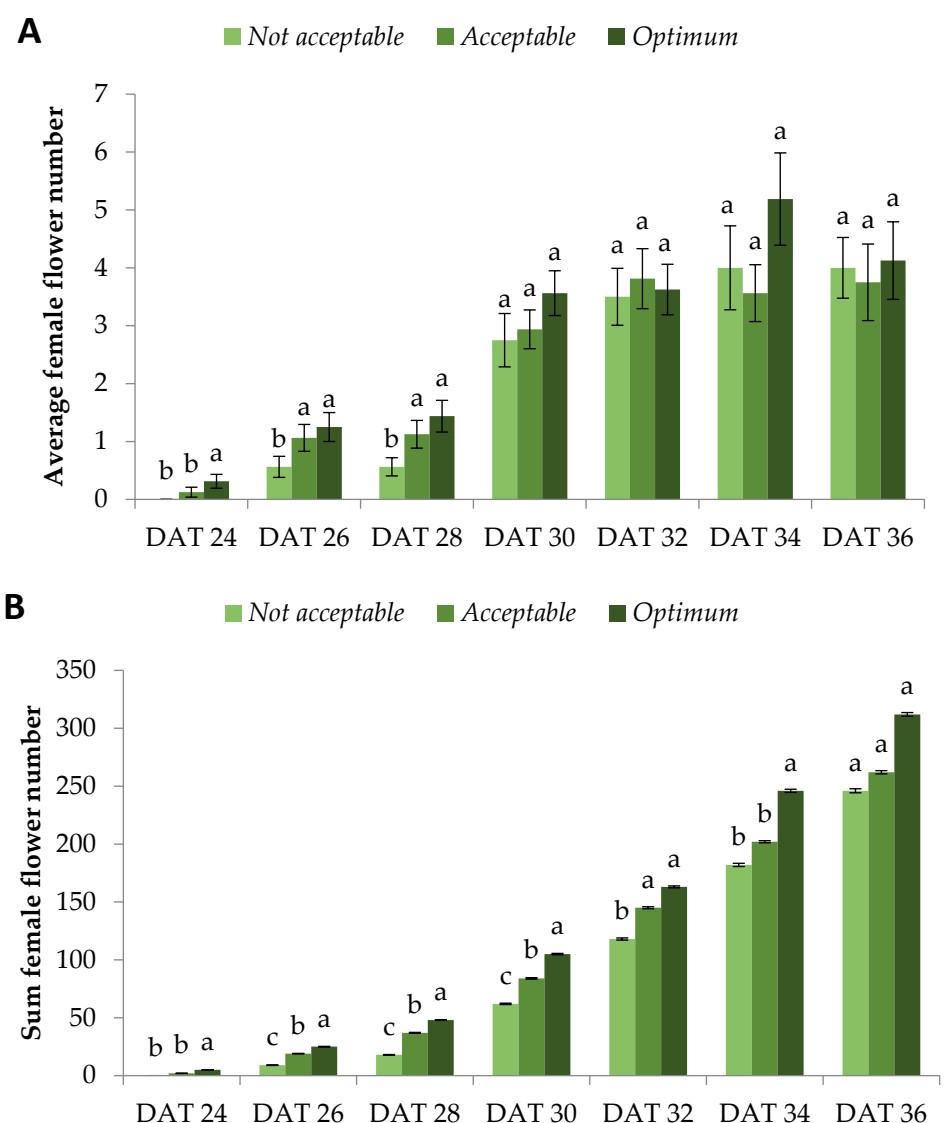

**Figure 2.** (**A**) Mean and (**B**) sum female flower number of watermelon plants from different quality categories at 24–36 days after transplanting (DAT). Within a DAT, average values (*n* = 8; ±SE) followed by different letters are significantly different (α < 0.05).

Subsequently, we evaluated the yield of fruits derived from flowers that bloomed on the specified days. This was achieved after meticulously labeling each female flower on the day it bloomed. The fruits were harvested 40 days after flowering, which is typical for watermelon crops cultivated in the Mediterranean region in the summer. To begin with, flowers that bloomed on DAT 24 did not lead to produced fruits in any quality category. *Optimum* and *Acceptable* plants bore fruits from flowers that bloomed on DAT 26, whereas *Not Acceptable* plants bore fruits from flowers that bloomed on DAT 30, four days later. *Optimum* plants had significantly greater sum yield compared with the other quality categories from DAT 28 up to DAT 32. *Acceptable* and *Not Acceptable* caught up with *Optimum* from DAT 34 and onwards, but a strong tendency towards the latter ensued (Figure 3A). An identical trend was exhibited for the sum fruit number, proving that the individual fruit weight was similar for all quality categories (Figure 3B). Indeed, this observation was expected since the fruit size and quality are genotypic features and cannot be considerably altered within the same environmental and growth conditions [24,25].

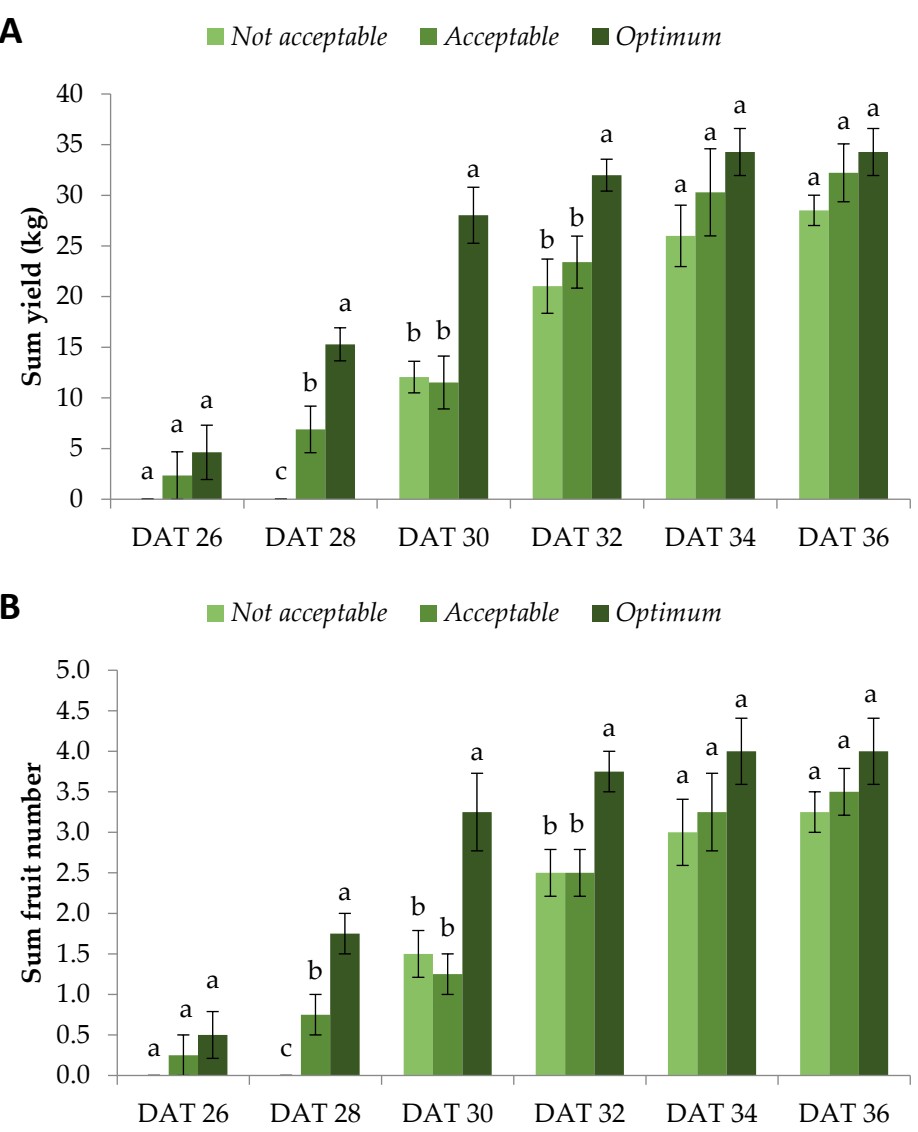

**Figure 3.** (**A**) Sum fruit yield and (**B**) sum fruit number of watermelon plants from different quality categories at 26–36 days after transplanting (DAT). Within a DAT, average values (*n* = 4; ±SE) followed by different letters are significantly different ($\alpha < 0.05$).

Strikingly, *Optimum* plants reached 82%, 93%, and 100% of their total yield from flowers that bloomed on DAT 30, DAT 32, and DAT 34, compared to DAT 36, respectively. The respective percentages for *Acceptable* plants were 36%, 73%, and 94%. The respective percentages for *Not Acceptable* plants were 42%, 74%, and 91%. Similar values were observed for fruit number. From these results, it is deduced that the yield of *Optimum* plants reached its peak considerably earlier compared with the other quality categories. Specifically, it is worth mentioning that the yield and fruit number of *Optimum* plants on DAT 30 were comparable to the *Acceptable* and *Not Acceptable* ones on DAT 34 and DAT 36, respectively, showing a four-day and six-day earlier crop production. This is particularly important, since farmers aim to harvest their products in the shortest possible time in order to enter the market earlier than their competition to achieve higher prices, as well as to limit their production costs.

Finally, the harvested fruits were evaluated with respect to their morphological and biochemical quality. As expected, no significant differences were observed in attributes such as the rind thickness and the fruit length and width. Moreover, important fruit phytochemicals regarded as nutritive molecules such as antioxidants (displayed by FRAP assay), phenolics, carotenoids, lycopene, and total soluble solids also did not exhibit

significant differences (Table 1). These results are not surprising, since fruit quality is known to be highly correlated with parameters such as the genotype, the environmental conditions, the soil properties, the growing season, the harvest stage, as well as the postharvest handling. All these parameters were identical in the quality categories of our experiment.

**Table 1.** Fruit biochemical and morphological parameters after cultivation of watermelon plants from different seedling quality categories. Within a row, average values ($n = 3$, ±SE) followed by different letters are significantly different ($\alpha < 0.05$).

| Parameters | Quality Categories | | |
| --- | --- | --- | --- |
| | *Not Acceptable* | *Acceptable* | *Optimum* |
| Rind thickness (cm) | 0.83 ± 0.09 a | 0.90 ± 0.12 a | 1.20 ± 0.17 a |
| Fruit length (cm) | 28.00 ± 1.89 a | 29.33 ± 0.93 a | 31.67 ± 0.93 a |
| Fruit width (cm) | 21.83 ± 0.60 a | 22.50 ± 0.58 a | 22.67 ± 0.73 a |
| FRAP (µg/g) | 80.86 ± 1.40 a | 75.80 ± 2.82 a | 78.88 ± 2.74 a |
| Phenolic content (mg/g) | 0.19 ± 0.01 a | 0.17 ± 0.01 a | 0.16 ± 0.01 a |
| Carotenoid content (µg/g) | 24.68 ± 3.30 a | 27.25 ± 1.51 a | 19.42 ± 2.20 a |
| Lycopene content (µg/g) | 18.35 ± 2.64 a | 19.60 ± 1.67 a | 14.37 ± 1.05 a |
| Total soluble solids (°Brix) | 11.23 ± 0.22 a | 11.60 ± 0.32 a | 11.10 ± 0.31 a |

It should be mentioned that the summer of 2021 was exceptionally hot, possibly provoking the crops to synthesize a great amount of photoprotective phytochemical compounds (i.e., phenolics, carotenoids, and antioxidants in general). For example, lycopene is a carotenoid with high antioxidant activity mainly present in watermelon and tomato and responsible for their red coloration. This compound plays a significant role against prostate cancer [26]. In another experiment [27], watermelon fruits in Greece, in 2018, which had milder summer temperatures, showed lower total phenolic content and antioxidant activity compared those in the present study. This was also observed in another study of our group which resulted in greater fruit quality in 2021 crops compared with that of 2018 crops [28]. The potentially enhanced fruit quality may lead to increased market value when yield is not negatively affected. However, this aspect should further be addressed in the coming years since harsh environmental conditions are already visible due to climate change. In addition, European watermelon is mainly produced in the Mediterranean region, which is a strong climate change hotspot.

Yield earliness is an important aspect of vegetables crops produced and consumed in certain seasons and for a limited period of time, such as watermelon. By achieving yield earliness of a few weeks, or even days, the producers and the stakeholders in general can enjoy higher prices due to the high market demand before the product enters the market. For example, watermelon farmers in Greece usually have a turnover of EUR 0.50–0.60 and EUR 0.10–0.20 per kg of early and late watermelon, respectively (personal communication). Temperature and sunlight are two crucial environmental factors affecting yield earliness. In Greece and other watermelon-producing countries, the crop is established in the field from late winter to early spring, when temperatures are still very low for such heat-requiring species. In order to overcome this, soil covers are used to increase the soil temperature (along with other benefits), while low tunnels are used to increase the daytime air temperature and reduce the heat losses in the nighttime.

According to the above, even a four- or six-day earlier crop harvest achieved by using higher-quality seedlings is important to enjoy the highest possible income. Therefore, nurseries should improve their techniques and equipment with the aim of producing higher percentages of *Optimum* seedlings and lower percentages of *Not acceptable* ones. With a view to achieving yield earliness, the producers should avoid the use of grafted watermelon seedlings which were either self-produced or produced by low-tech non-commercial nurseries. They should only use seedlings from high-tech commercial nurseries which provide certified transplanting material of the highest possible quality according to

their information. Obviously, such seedlings have higher cost, but this should comfortably be overcome by the price difference of early-marketed fruits.

## 4. Conclusions

For many years, agronomists, nurseries, and farmers have sought a recipe for producing high-quality vegetable seedlings to be established and cultivated in the field with limited transplantation repercussions. Until recently, the quality of grafted watermelon seedlings was only objectively defined through general morphological characteristics. Moreover, the seedling quality has never been compared with their respective transplantation shock plants and fruit productions after a full growth cycle in the field, meaning that seedlings were categorized in quality groups in a superficial manner. Here, we report that the seedling quality before transplantation is crucial for the subsequent vegetative, floral, and fruit development in the field. Plants derived from seedlings labeled as *Optimum* outperformed the *Acceptable* and *Not Acceptable* counterparts in growth parameters such as stem diameter, plant area, and leaf number up to the flowering initiation. The same quality category flowered earlier and at a more rapid pace. Most importantly, *Optimum* plants developed ripe fruits four days faster than the *Acceptable* ones and six days faster than the *Not Acceptable* ones, showing considerable yield earliness. This is particularly important in order for the producers and the stakeholders in general to enjoy higher prices. As expected, fruit morphology and quality were similar in all quality categories since these parameters are mainly defined by genotypic characteristics. Our findings are particularly important for nurseries and farmers in order to schedule their production and select the most appropriate seedlings to limit the transplantation shock and enhance the development of their crops. Obviously, such seedlings have higher cost, but this should comfortably be overcome by the price difference of early-marketed fruits.

**Author Contributions:** Conceptualization, methodology, and data analysis: F.B. and A.K.; experimental measurements: F.B.; writing—original draft preparation: F.B.; writing—review and editing: F.B. and A.K.; supervision and project administration: A.K. All authors have read and agreed to the published version of the manuscript.

**Funding:** This research was co-financed by the European Union and Greek national funds through the Operational Program Competitiveness, Entrepreneurship and Innovation, under the call RESEARCH–CREATE–INNOVATE (project code: T1EDK-00960, LEDWAR.gr).

**Data Availability Statement:** Data sharing is not applicable to this article.

**Acknowledgments:** The authors would like to express their gratitude to Christodoulos Dangitsis, Eleni Papoui, Emmanouil Kokolakis, and Anna Gkotzamani, Agronomists, for their assistance with the experimental measurements, as well as to Kalliopi Radoglou, who critically reviewed the manuscript and commented on its English.

**Conflicts of Interest:** The authors declare no conflict of interest.

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
