# Peer review of "The Use of High-Quality Watermelon Seedlings Is Prerequisite to Limit the Transplanting Shock and Achieve Yield Earliness"

_horticulturae, doi:10.3390/horticulturae9080943_

Round 1

Reviewer 1 Report

I appreciate the invitation to review this study, which considers the importance of seedling quality in post-transplant agronomic performance to the field.
The manuscript is very well written. I made few considerations which are present in the manuscript, but I take the opportunity to transcribe them here. I emphasize that even the question I raised about the test of means is not accepted by the authors, the maintenance of the Scott-Knott test does not make the approval of the manuscript for publication inappropriate.

My observations are:

Line 15: brackets

Line 54: a new paragraph

Lines 78-83: Making the issue clearer

Line 85: lowercase letter

Line 124: first time it is interesting to define

Line 150: conducted

Line 166: I understand that the better test to use in this case is Tukey because there are few treatents. In many results shown in figures and table there may be differences. But I accept the authors' decision.

Author Response

I appreciate the invitation to review this study, which considers the importance of seedling quality in post-transplant agronomic performance to the field.

The manuscript is very well written. I made few considerations which are present in the manuscript, but I take the opportunity to transcribe them here. I emphasize that even the question I raised about the test of means is not accepted by the authors, the maintenance of the Scott-Knott test does not make the approval of the manuscript for publication inappropriate.

Response: The authors would like to express their gratitude to the reviewers for the time they invested for assessing our manuscript and their useful comments and suggestions. The comments were responded one-by-one below. Please note that our responses were given according to the pdf file, since the comments from the Horticulturae system and the pdf were practically the same.

My observations are:

Line 15: Please delete one of the brackets.

Response, L15: Corrected. Thank you for the observation.

Line 54: I think starting a new paragraph with this sentece makes the text better,

Response, L58: A new paragraph started according to your suggestion.

Lines 78-83: Previously, you have correctly built the justification of the importance of the quality of the seedling to be grafted, in order to gain in operational performance of nurseriers and reduce costs. Here in the hypothesis it seems that the evaluation will be done on "high quality seedlings", "acceptable" and "not acceptable", but already grafted. Am I correct? If yes, then make this clear to the reader.

Response, L84-87: This part was rewritten for better clarification according to your suggestion.

Line 85: lowercase letter

Response, L92:  Corrected. Thank you for the observation.

Line 124: Please: days after transplantation (DAT)

Response, L143:  Corrected. Thank you for the observation.

Line 150: conducted

Response, L188: Corrected. Thank you for the observation.

Line 166: I understand that the better test to use in this case is Tukey because there are few treatents. In many results shown in figures and table there may be differences. But I accept the authors' decision.

Line 166, Comments from pdf: I think this method is more applicable when there are many treatments. Maybe, Tukey's test could show difference between the treatments.

Line 267, Figure 3. Again, I consider the Tukey test more appropriate than the Scott-Knott test for this number of treatments.

Please see DAT 26, 34 and 36 for both Figure 3A and 3B

Line 283, Table 1. Again, I consider the Tukey test more appropriate than the Scott-Knott test for this number of treatments

Response, L203-206: Thank you for your suggestion. The Scott-Knott method presents important and unique characteristic by which the grouping results do not overlap. This is critical to obtain values that separate the three quality categories without therefore overlapping results between them. It is now mentioned in the text along with another reference.

We also tested our data using Tukey post-hoc. In Figure 3A and 3B, DAT 26, 34 and 36 showed the same results when tested with Scott Knott and Tukey methods. The same applies for Table 1 as well.

Reviewer 2 Report

See the comments on the manuscript.

Minor editing of English language required

Author Response

Note from the authors. The authors would like to express their gratitude to the reviewer for the time they invested for assessing our manuscript and their useful comments and suggestions. The comments were responded one-by-one below.

Introduction: The introduction should provide a clear and concise overview of the problem being addressed, including the significance and relevance of the research question, and a brief literature review that highlights the gap in knowledge that the study aims to address.

Response: The introduction was improved as suggested. Specifically, a comment was included about the importance of high-quality seedlings, while the objectives were rewritten for better clarification. Please see L40-41 and L84-87.

Methods: The methods section should provide detailed information on the methodology used to conduct the study, including the experimental design, data collection, and data analysis techniques

Response: The materials and methods section was improved as suggested. Specifically, the acclimatization stage was thoroughly described, while the seedlings’ evaluation for quality grouping was explained in detail.  Measurements such as stem diameter and plant area were described better. In addition, the rationale behind the harvest time and fruit weight for biochemical analyses, was included, as well as some information about Scott-Knott statistical analysis. Please see L99-103, L107-109, L115-120, L143-152, L165-170, L177-181, and L203-206.

The results of the method are clearly presented, but it would be helpful to provide more context around the significance of the improvements in average precision and accuracy over previous methods.

Response: The results and discussion section was improved as suggested. Specifically, we enhanced the discussion about transplantation shock and the implications to seedlings from each quality category. In addition, we further discussed the fruit characteristics, the possible market value, as well as the potential reasons for the lack of significant differences on fruit quality. Finally, we added practical implications of our results, including economic benefits and practical recommendations. Please see L273-286, L331-335, and L338-371.

Conclusion: The conclusion section should provide a brief summary of the study's findings and their significance, as well as any practical applications of the research. It should also restate the study's objectives and indicate whether they were achieved.

Response, L: The conclusion was improved according to your suggestions. Please see L390-391 and L396-397.

Reviewer 3 Report

Overall, this research article represents an interesting investigation on “The use of high-quality watermelon seedlings is prerequisite to limit the transplanting shock and achieve yield earliness.” The manuscript presents an intriguing study that investigates the impact of seedling quality on the growth, flowering, and fruit development of watermelon plants. The results demonstrate a clear correlation between seedling quality and subsequent plant performance. However, a few points need to be addressed before the manuscript can be considered for publication. Below are some specific comments for the authors' consideration:

The manuscript introduces seedling quality categories as "Optimum," "Acceptable," and "Not Acceptable." The description of the criteria used to categorize seedlings into "Optimum," "Acceptable," and "Not Acceptable" categories lacks clarity. It's essential to provide explicit details about how the macroscopic evaluation of leaf area, color, root system, and seedling vigor translated into these categorizations. This would enhance the readers' understanding of the selection process and the reliability of the categorizations. Additionally, the methodology section should include detailed information about how seedlings were assessed and categorized.

The manuscript mentions that "Not Acceptable" seedlings exhibited delayed flowering compared to "Optimum" ones. It would be insightful to discuss the potential reasons behind this delay—whether it's related to physiological factors, stress, or other environmental influences.

The manuscript mentions a 14-day acclimatization phase for the seedlings in a plastic Venlo-type greenhouse, but it would be helpful to provide more information about the specific conditions and care during this phase. Details such as temperature, humidity, light exposure, and irrigation practices could influence the subsequent performance of the seedlings.

While the measurement parameters like stem diameter, plant area, and leaf number are mentioned, the manuscript lacks a clear explanation of the measurement procedures. Providing a step-by-step description of how these measurements were conducted and any specific tools used (e.g., caliber, WinRHIZO Pro software) would ensure the replicability of the study.

The timing of fruit harvest and subsequent analysis is described, but some points could be clarified. For instance, the rationale behind waiting 40 days from each flowering date before harvesting should be explained. Additionally, the specific reason for selecting fruits of approximately 8 kg for biochemical analysis could be discussed.

While the study indicates that seedling quality did not significantly affect photosynthetic mechanism and fruit morphology, these findings warrant a more comprehensive discussion. The authors should delve into the implications of these results, considering their potential impact on long-term plant health, overall fruit quality, and market value. It would be helpful to speculate on the reasons for the lack of significant differences in these aspects.

The manuscript could benefit from a more extensive discussion on the practical implications of the findings. How can farmers and nurseries adapt their practices based on the study's results? What are the potential economic benefits of using "Optimum" seedlings in terms of yield earliness? Providing specific recommendations and strategies for implementation would enhance the manuscript's applied value.

The manuscript is generally well-written, but some sentences could be rephrased for clarity and coherence. Additionally, there are a few instances where the wording is somewhat convoluted. A thorough proofreading for language and syntax is recommended.

This manuscript makes a valuable contribution to our understanding of the impact of seedling quality on watermelon plant performance. Addressing the points raised above would significantly improve the clarity and rigor of the study. With these revisions, the manuscript could be suitable for publication in Horticulturae.

The manuscript is generally well-written, but some sentences could be rephrased for clarity and coherence. Additionally, there are a few instances where the wording is somewhat convoluted. A thorough proofreading for language and syntax is recommended.

Author Response

Overall, this research article represents an interesting investigation on “The use of high-quality watermelon seedlings is prerequisite to limit the transplanting shock and achieve yield earliness.” The manuscript presents an intriguing study that investigates the impact of seedling quality on the growth, flowering, and fruit development of watermelon plants. The results demonstrate a clear correlation between seedling quality and subsequent plant performance. However, a few points need to be addressed before the manuscript can be considered for publication. Below are some specific comments for the authors' consideration:

Response: The authors would like to express their gratitude to the reviewer for the time they invested for assessing our manuscript and their useful comments and suggestions. We would like to let you know that we enjoyed your fruitful comments which ultimately improved our manuscript. The comments were responded one-by-one below.

The manuscript introduces seedling quality categories as "Optimum," "Acceptable," and "Not Acceptable." The description of the criteria used to categorize seedlings into "Optimum," "Acceptable," and "Not Acceptable" categories lacks clarity. It's essential to provide explicit details about how the macroscopic evaluation of leaf area, color, root system, and seedling vigor translated into these categorizations. This would enhance the readers' understanding of the selection process and the reliability of the categorizations. Additionally, the methodology section should include detailed information about how seedlings were assessed and categorized.

Response: More information was added about the quality grouping as suggested. Please see L107-110 and L115-120.

The manuscript mentions that "Not Acceptable" seedlings exhibited delayed flowering compared to "Optimum" ones. It would be insightful to discuss the potential reasons behind this delay—whether it's related to physiological factors, stress, or other environmental influences.

Response: The potential reasons behind flowering delay of "Not Acceptable" seedlings was further discussed in the text, as suggested. Please see L273-286.

The manuscript mentions a 14-day acclimatization phase for the seedlings in a plastic Venlo-type greenhouse, but it would be helpful to provide more information about the specific conditions and care during this phase. Details such as temperature, humidity, light exposure, and irrigation practices could influence the subsequent performance of the seedlings.

Response: More information was added about the acclimatization period as suggested. Please see L99-103.

While the measurement parameters like stem diameter, plant area, and leaf number are mentioned, the manuscript lacks a clear explanation of the measurement procedures. Providing a step-by-step description of how these measurements were conducted and any specific tools used (e.g., caliber, WinRHIZO Pro software) would ensure the replicability of the study.

Response: More information was added about the measurements as suggested. Please see L143-152.

The timing of fruit harvest and subsequent analysis is described, but some points could be clarified. For instance, the rationale behind waiting 40 days from each flowering date before harvesting should be explained. Additionally, the specific reason for selecting fruits of approximately 8 kg for biochemical analysis could be discussed.

Response: More information was added about the rationale behind the harvest time and the weight of the analyzed fruits, as suggested. Please see L165-170 and L177-181.

While the study indicates that seedling quality did not significantly affect photosynthetic mechanism and fruit morphology, these findings warrant a more comprehensive discussion. The authors should delve into the implications of these results, considering their potential impact on long-term plant health, overall fruit quality, and market value. It would be helpful to speculate on the reasons for the lack of significant differences in these aspects.

Response: The fruit characteristics, the possible market value, as well as the potential reasons for the lack of significant differences were further discussed in the text, as suggested. Please see L331-335 and L339-349.

The manuscript could benefit from a more extensive discussion on the practical implications of the findings. How can farmers and nurseries adapt their practices based on the study's results? What are the potential economic benefits of using "Optimum" seedlings in terms of yield earliness? Providing specific recommendations and strategies for implementation would enhance the manuscript's applied value.

Response: Practical implications, including economic benefits and practical recommendations were included in the manuscript as suggested. Please see L350-371.

The manuscript is generally well-written, but some sentences could be rephrased for clarity and coherence. Additionally, there are a few instances where the wording is somewhat convoluted. A thorough proofreading for language and syntax is recommended.

This manuscript makes a valuable contribution to our understanding of the impact of seedling quality on watermelon plant performance. Addressing the points raised above would significantly improve the clarity and rigor of the study. With these revisions, the manuscript could be suitable for publication in Horticulturae.

Comments on the Quality of English Language

The manuscript is generally well-written, but some sentences could be rephrased for clarity and coherence. Additionally, there are a few instances where the wording is somewhat convoluted. A thorough proofreading for language and syntax is recommended.

Response: The manuscript and its English was critically reviewed by Professor Kalliopi Radoglou who is acknowledged in the respective section. Please see L407-408.

Round 2

Reviewer 2 Report

Accept in present form.

Good luck!

Minor editing of English language required